# Size-Mediated Trophic Interactions in Two Syntopic Forest Salamanders

**DOI:** 10.3390/ani13081281

**Published:** 2023-04-07

**Authors:** Andrea Costa, Giacomo Rosa, Sebastiano Salvidio

**Affiliations:** Department of Earth, Environment and Life Sciences (DISTAV), University of Genova, Corso Europa 26, 16132 Genova, Italy; giacomorosa@live.it (G.R.); sebastiano.salvidio@unige.it (S.S.)

**Keywords:** competition, foraging, diet, salamander, species interaction, interference competition

## Abstract

**Simple Summary:**

Organisms compete for resources, such as food and space, and competition can occur in two ways: exploitative competition or interference competition. In exploitative competition, organisms consume the same limited resources, directly reducing the availability of those resources for other individuals, while in the case of interference competition, organisms actively prevent others from accessing resources, independently of resource availability. We tested for the presence and type of foraging competition in two species of forest-dwelling salamanders in Italy: *Speleomantes strinatii* and *Salamandrina perspicillata*. Our results suggested the presence of an interference/interaction occurring between the two species and affecting the foraging activity of the smaller one (*Salamandrina perspicillata*). This competitive interaction is size mediated and configured as interference competition rather than exploitative competition.

**Abstract:**

Exploitative competition and interference competition differ in the way they affect re-source availability for competitors: in the former, organisms reduce resource availability for the competitors; in the latter, one organism actively prevents the competitor from accessing resources, independently of their availability. Our aim is to test for the presence of foraging competition in two forest-dwelling salamanders in Italy: *Speleomantes strinatii* and *Salamandrina perspicillata*. We also aim at testing for size-mediated competition. We obtained stomach contents from 191 sampled individuals by means of stomach flushing at 8 sampling sites where both species occur. We focused our analysis on the core prey taxa shared by both species: Collembola and Acarina. We found that the foraging activity of *S. perspicillata* is positively affected by body size and negatively affected by potential competitor’s activity on the forest floor during the sampling, which also significantly weakened the positive relationship with body size. These results suggest the presence of an interference/interaction occurring between the two species and affecting the foraging activity of *S. perspicillata*. This competitive interaction is size mediated and configured as interference competition rather than exploitative competition.

## 1. Introduction

Competition is a fundamental aspect of ecological systems, shaping the distribution and abundance of organisms, and it plays a crucial role in determining which species will survive and thrive in a particular environment [1,2]. Two types of competition, i.e., exploitative and interference, differ in the way they affect resource availability for competitors. Exploitative competition occurs when organisms consume the same limited resources, directly reducing the availability of those resources for other individuals [2]. Interference competition, by contrast, occurs when one organism actively prevents or hinders another from accessing resources [3], independently of resource availability. Both types of competition can have significant effects on population dynamics and community structure [4]. Exploitative competition can lead to the exclusion of certain species from a particular habitat or trigger resource partitioning among several niche axes, while interference competition can both lead to exclusion (e.g., [5]) or changes in the behavior or in the way species access resources [3].

Foraging, i.e., the way in which food and energy are actively obtained and maximized from a pool of resources available in the environment, is one of the most important activities carried out by animals and has multiple ecological and life history drawbacks [6]. Indeed, foraging activity may affect and modulate for individual [7] or population survival [8], reproductive fitness [9,10], character displacement [11], and inter-specific [12,13,14] or intra-specific [15,16] resource partitioning and competition. In particular, the degree of overlap in diet composition is responsible for modulating the strength of competition both between species and within populations [17,18]. Moreover, dietary composition, and in particular niche breath plasticity, can impact species’ or populations’ ability to withstand environmental changes and face extinction risk, with specialized species being more prone to extinction, given their supposed inability to expand the trophic niche [19,20]. Populations’ and communities’ foraging activity also has major implications on interspecific interactions and ecosystem structure and functioning, spanning from trophic cascades to nutrient cycling (e.g., [21,22]). For all these reasons, the study of diet composition and foraging behavior of a given organism, or biological community, is of pivotal importance in ecology and also encompasses conservation implications [23,24,25].

Amphibians represent the most endangered vertebrate group worldwide, with habitat loss and alteration being one of the main drivers of their global decline [26]. Studying the trophic ecology of amphibian populations and how it relates to environmental changes, or to interspecific interactions, is of paramount importance to better understanding ecological and demographic processes related to population persistence. Moreover, for this reason, amphibians in general and salamanders (Urodela) in particular have been widely adopted as ecological models for studying environmental changes and testing ecological theories [27]. Salamanders inhabiting temperate ecosystems have been studied intensively to disclose their trophic ecology, foraging strategies, and prey–predator interactions (e.g., [28,29]). Indeed, all the almost 800 extant species of salamanders are obligate carnivores, and in temperate forest ecosystems they act as top predators on the forest floor [30]. Given the high density and biomass they can reach [31], salamanders are supposed to both regulate invertebrates’ abundance and affect litter decomposition rate [32]. Salamanders, being ectotherms, are particularly efficient in converting food intake biomass [33], and since they are consumed by larger vertebrates, they are also particularly efficient in converting invertebrate to vertebrate biomass and providing energy transfer to higher trophic levels [33].

The principal aim of this paper is to test for the presence of foraging competition in two forest-dwelling salamanders inhabiting the Northern Apennine Mountain chain of Italy: the Strinati’s cave salamander, *Speleomantes strinatii,* and the Northern spectacled salamander, *Salamandrina perspicillata*. Within the study area, the two species are largely sintopic and inhabit the same forest microhabitats, but differ in body size, population trophic strategy [29,34], and the response of the foraging activity to local weather conditions [35]. In this study, we aim at testing for size-mediated competition by testing the foraging activity of the two salamanders in eight sampling sites where both species occur. In particular, we will evaluate the following: (i) if the foraging activity on the shared prey categories is influenced by the potential competitor activity on the forest floor; (ii) if the potential competition is modulated by body size; and (iii) if the observed interaction, if any, can be attributed to exploitative or interference competition. We expect that, if competition is present, we should observe a significant reduction in use for a given shared resource when both species are active and forage on the forest floor.

## 2. Materials and Methods

### 2.1. Study Area

Salamanders were sampled in eight sampling sites across a homogenous area of the Northern Apennines of Italy (Figure 1), covering approximately 60 km^2^. The study area is characterized by a sub-Mediterranean climate, with annual rainfall comprising between 1600 and 2100 mm and usually displaying two rainfall peaks per year: the first in spring (April–May) and the second in autumn (October–November) [36]. In all the sampling sites, both *Speleomantes strinatii* and *Salamandrina perspicillata* are present, but are active on the forest floor and forage under different climatic conditions [35].

### 2.2. Study Species

Strinati’s cave salamander, belonging to the family Plethodontidae, has a maximum length of about 125 mm, while the Northern spectacled salamander, belonging to the family Salamandridae, does not exceed 100 mm in total length [37]. *Speleomantes strinatii* is a fully terrestrial species capable of establishing stable populations both in surface and underground environments [28,38,39] and displays direct development. *Salamandrina perspicillata*, despite having a generally terrestrial activity for the majority of the year [40], displays a biphasic life cycle and only reproductive females enter water for egg deposition [37]. Adults of both species are lungless and usually active on the forest floor after periods of rainfall [34,40,41]. In forests, the diet of *S. strinatii* and *S. perspicillata* has been extensively described, both in the spring and autumn seasons, by means of stomach flushing [34,42,43], a non-lethal and robust technique [44]. Both species feed on a large variety of invertebrate prey but, at the population level, *S. strinatii* behaves as a generalist while *S. perspicillata* is more specialized on smaller taxa, such as springtails and mites [34,43]. Although some minor differences exists in the diet composition of the two species between the spring and autumn seasons, the main foraging pattern remains unchanged during the year [34,43]. The two species also display slightly different activity patterns for foraging, with *S. perspicillata* being more active with higher air temperatures and requiring higher levels of moisture in the forest floor [35].

### 2.3. Salamanders’ Diet Sampling

Salamanders were captured at eight sites, during the day, immediately after rainfall, while active on the forest floor. Sampling occurred in the spring of 2021, from April 22nd to May 20th (Table 1), a period corresponding to the peak of activity for both species [37]. Captured salamanders were stomach flushed in the field, and their snout-vent length (SVL) was measured by means of a plastic ruler [34]. Stomach contents were preserved in 70% ethanol and analyzed in the lab under dissecting microscopes, identifying prey items at the lowest possible taxonomic level [16,34]. Since some sites were visited on different days within the sampling season, we employed photographic identification to avoid pseudo-replication, a technique successfully employed for *S. perspicillata* [45] and *Speleomantes* [46].

### 2.4. Data Analysis

In order to evaluate the potential presence of behavioral interactions between *S. strinatii* and *S. perspicillata,* we focused on the shared prey items that constitute the core diet of the two species, i.e., the prey categories that are consumed by an *f* fraction of the individuals (sensu [47]). Thus, we calculated the frequency of occurrence of each prey item in the stomachs of the sampled salamanders (*f*) and retained for subsequent analyses only the prey items, shared by both predator species, that accounted for *f* > 0.1, i.e., prey items that were present in the diet of >10% of both Strinati’s cave salamanders and spectacled salamanders. Therefore, the dataset used for the analysis consists, for each individual salamander, of the number of prey items for each of the shared prey taxa with *f* > 0.1 for both predators, the SVL of each individual salamander, and a categorical variable (0; 1) describing if the potential competitor salamander was active on the forest floor when sampling occurred. We built a series of generalized linear models (GLMs) for both *S. perspicillata* and *S. strinatii*; one model was built for each shared prey category in the core diet, accounting for a Poisson error distribution and log-link function, assuming the number of prey items per stomach as the response variable, and SVL and competitor presence as predictors. Moreover, in order to test if resource use for a given prey taxon is mediated by a size-dependent interaction between the two species, we also included an interaction term between SVL and competitor presence. Prior to model building, we scaled continuous predictors (SVL) by means of R function scale (), which subtracts the sample mean from each value and divides it by the SD. Model adequacy was assessed through visual inspection of the residuals, and significance levels for the analysis were set at alpha = 0.05. All analyses were conducted within the R (R Core team) environment.

## 3. Results

During sampling, we found *S. perspicillata* active on the forest floor in five out of eight sampling sites and *S. strinatii* active in six out of eight sampling sites, while we found both species simultaneously foraging in three out of eight sampling sites. From sampling, we obtained stomach contents from 191 individual salamanders: 85 *S. perspicillata* and 106 *S. strinatii,* and the sample size for each species at each sampling site is reported in Table 1. From stomach contents sorting, we determined a total of 1360 prey items, divided into 28 prey categories (i.e., prey taxa). The number of prey items per stomach ranged between 1 and 28 (mean = 7.09; SD = 5.70), and the complete diet data for the 2 species are reported in Table 2. We found that the core diet (*f* > 0.1) of the two sampled species consisted of four taxa for *S. strinatii* (Acarina, Collembola, Diptera larvae, and Formicidae) and two taxa for *S. perspicillata* (Acarina and Collembola). Only 2 prey categories with *f* > 0.1 were shared by the 2 species: Springtails (Collembola) and Mites (Acarina); therefore, we built models for each species based on these 2 prey taxa.

Number = number of items for a given prey category; n of stomachs = number of individual salamanders in which a given prey category has been found; f = frequency of occurrence for a given prey taxon.

From the results of the GLMs (complete results are reported in Table 3), *S. perpsicillata*’s consumption of both Collembola and Acarina is negatively affected by potential competitor’s activity on the forest floor during the sampling (*β*-Collembola = −0.36, *p* < 0.05; *β*-Acarina = −1.84, *p* < 0.001).

At the same time, the number of both Collembola and Acarina per stomach were positively affected by body size (*β*-Collembola = 0.67, *p* < 0.001; *β*-Acarina = 0.43, *p* < 0.001), while the interaction between competitor’s presence and body size resulted negative and significant for both prey categories (Figure 2; Table 3). Concerning *S. strinatii*, SVL had a negative significant effect on the use of Collembola (*β* = −0.82, *p* < 0.001), but not on Acarina (*β* = −0.44, *p* = 0.11), while neither competitor’s activity nor the interaction between body size and competitor’s presence had a significant effect on prey use (Figure 2; Table 3).

## 4. Discussion

Among the many resources for which organisms compete, food resources are the most important. Although the two studied species share many trophic resources, no evidence of competition has been observed so far, even when tested for trophic niche overlap on the total realized niche [34]. In the present study, we confirmed that the spectacled salamander and Strinati’s cave salamander are both capable of including a large variety of prey items in their diet. This trait has already been observed regardless of whether the two species occur in syntopy [34], or not [42,43,48,49]. However, only a small fraction of the realized trophic niche of both species is comprised within the core diet *f* > 0.1, which implies that a large number of individuals share few common resources, while also including other prey in the diet at a lower frequency of occurrence. This pattern of individual variation, for both species, has already been observed and studied in detail within the framework of individual specialization (e.g., [29,43]).

The main results of the study, however, deal with the focus on the two shared prey categories in the core diet (*f* > 0.1) of both species: Acarina and Collembola. Indeed, our study showed that the foraging activity of the spectacled salamander is negatively affected by the concurrent activity of the Strinatii’s cave salamander on the forest floor, and not vice versa. This pattern suggests the existence of some sort of competitive interaction between the two study species. Indeed, *S. perspicillata* consumed more prey items of the two core diet categories when *S. strinatii* was not actively foraging. Moreover, we found body size to positively affect the consumption of both prey categories only in *S. perspicillata*, in particular when *S. strinatii* was not active on the forest floor. This latter result suggests a size-mediated interaction between the two species. In this respect, Salvidio et al. [34], testing trophic niche overlap between *S. strinatii* and *S. perspicillata*, found a weak overlap among adults of the two species; however, some sort of overlap exists between *S. strinatii* subadults and *S. perspicillata* adults, which have similar body size. In addition, we also observed that, while for *S. perpicillata* the two considered prey categories with *f* > 0.1 are numerically predominant in the realized trophic niche (Table 2), this is not the case for *S. strinatii*, which consumes these preys in much lower quantities, suggesting that the shared invertebrate preys are not limited resources. This pattern, where the activity of one species (*S. strinatii* in our case) has a direct effect on the resource use of another species (*S. perspicillata*), but not by reducing resource availability, is consistent with the theoretical framework of interference competition, rather than with exploitative competition (e.g., [3]).

With the present study, we provided evidence about an interference/interaction occurring between the spectacled salamander and Strinati’s cave salamander and affecting their foraging activity. As we can ascertain from the present study, this competitive interaction is, at least to some extent, size-mediated, and is configured as interference competition rather than exploitative competition, at least concerning trophic resources. The results of the present study also corroborate the observations of Rosa et al. [35] about the different foraging activity of *S. strinatii* and *S. perspicillata* in relation to local weather conditions. In the light of our new findings, the different foraging activity of these species in relation to precipitation and temperature [35] could also be interpreted as competitor’s avoidance or temporal niche partitioning [50,51]. Future studies and research development should be directed towards better understanding of the existing interactions between these two species of salamanders in order to ascertain whether the observed interaction on the trophic niche is also present on other niche axes, and whether this interaction has any direct effect on local abundance or density.

Finally, although the foraging strategy of the two study species is known to be consistent across seasons [34,43], our results were obtained from a cross-sectional study conducted only during spring. For this reason, we also suggest that future studies on this topic should consider employing longitudinal diet data, instead of cross-sectional ones, in order to better disclose the magnitude of the interaction described and how it could be modulated by seasonal effects, such as climatic conditions or resource availability.

## Figures and Tables

**Figure 1 animals-13-01281-f001:**
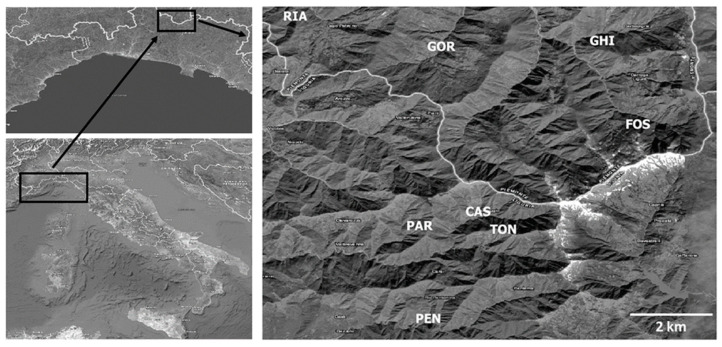
Map depicting the location of the eight sampling sites in Northern Italy. Site IDs on the right box correspond to Table 1, where additional details on site municipality and sample sizes are provided. Precise locations of sampling sites are not provided, to avoid poaching and pathogen spread.

**Figure 2 animals-13-01281-f002:**
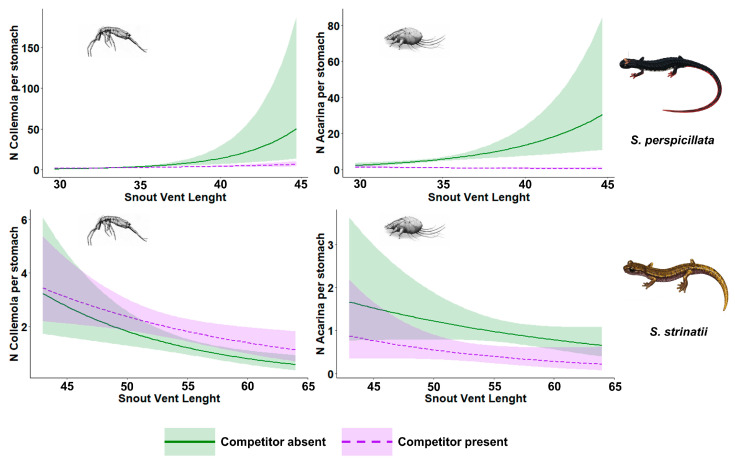
Plot representing the effect of body size (SVL) on the consumption of Collembola and Acarina for both *S. perspicillata* and *S. strinatii* when the potential competitor is present (purple) or absent (green) on the forest floor, as estimated by Poisson GLMs.

**Table 1 animals-13-01281-t001:** Details on sampling sites, sampled species for each site, and sample size for each species. Site ID corresponds to Figure 1. Precise locations of sampling sites are not provided to avoid poaching and pathogen spread.

Municipality	Site ID	Altitude Asl	Species	Sample Size
Valbrevenna (GE)	TON	789	*Salamandrina perspicillata*	11
			*Speleomantes strinatii*	18
	CAS	750	*Speleomantes strinatii*	20
	PAR	920	*Salamandrina perspicillata*	18
			*Speleomantes strinatii*	12
Carrega Ligure (AL)	GHI	760	*Salamandrina perspicillata*	29
			*Speleomantes strinatii*	14
	FOS	820	*Speleomantes strinatii*	23
Montoggio (GE)	PEN	580	*Speleomantes strinatii*	20
Cabella Ligure (AL)	GOR	620	*Salamandrina perspicillata*	13
Mongiardino Ligure (AL)	RIA	570	*Salamandrina perspicillata*	13

**Table 2 animals-13-01281-t002:** Number of items, divided by prey taxa, found in the stomach contents of 191 salamanders.

	*Salamandrina perspicillata* (n = 85)			*Speleomantes strinatii* (n = 106)		
Prey Taxon	Number	n of Stomachs	f	Number	n of Stomachs	f
Pseudoscorpiones	13	12	0.02	4	4	0.01
Acarina	178	47	0.30	76	37	0.10
Opilionida	4	4	0.01	13	10	0.02
Araneae	22	17	0.04	51	37	0.07
Diplopoda	10	9	0.02	65	39	0.08
Geophiloorpha	0	0	0.00	1	1	0.00
Litobiomorpha	0	0	0.00	8	6	0.01
Collembola	239	48	0.40	153	58	0.20
Diplura	1	1	0.00	0	0	0.00
Coleoptera	30	24	0.05	56	36	0.07
Coleoptera larvae	1	1	0.00	3	3	0.00
Diptera	14	9	0.02	20	17	0.03
Diptera larvae	12	11	0.02	123	36	0.16
Hemiptera	4	4	0.01	10	8	0.01
Heteroptera	2	2	0.00	6	5	0.01
Hemiptera larvae	1	1	0.00	0	0	0.00
Hymenoptera (winged)	2	2	0.00	8	6	0.01
Formicidae	12	8	0.02	116	44	0.15
Lepidoptera larvae	0	0	0.00	3	2	0.00
Neuroptera	1	1	0.00	2	2	0.00
Neuroptera larvae	0	0	0.00	3	3	0.00
Plecoptera	2	1	0.00	1	1	0.00
Trichoptera	1	1	0.00	1	1	0.00
Nematoda	0	0	0.00	2	2	0.00
Oligochaeta	1	1	0.00	20	5	0.03
Lumbricidae	0	0	0.00	2	1	0.00
Copepoda	0	0	0.00	1	1	0.00
Isopoda	43	14	0.07	19	13	0.02

**Table 3 animals-13-01281-t003:** Detailed results of the generalized linear models explaining the potential presence of competitive interactions on the prey items constituting the core diet of *S. perspicillata* and *S. strinatii*.

** *S. perspicillata* ** **Collembola**	**Parameter**	**Estimate**	**Std. Error**	***p*-Value**
	Intercept	1.30	0.11	-
	*β*–Competitor’s presence (1)	−0.36	0.14	*p* < 0.05 *
	*β*–SVL	0.67	0.16	*p* < 0.001 *
	*β*–SVL * Competitor’s presence	−0.44	0.18	*p* < 0.05 *
** *S. perspicillata* ** **Acarina**	**Parameter**	**Estimate**	**Std. Error**	***p*-value**
	Intercept	1.75	0.09	-
	*β*–Competitor’s presence (1)	−1.84	0.17	*p* < 0.001 *
	*β*–SVL	0.43	0.12	*p* < 0.001 *
	*β*–SVL * Competitor’s presence	−0.61	0.19	*p* < 0.01 *
** *S. strinatii* ** **Collembola**	**Parameter**	**Estimate**	**Std. Error**	***p*-value**
	Intercept	4.72	1.32	-
	*β*–Competitor’s presence (1)	−1.19	1.69	*p* = 0.48
	*β*–SVL	−0.82	0.23	*p* < 0.01 *
	*β*–SVL * Competitor’s presence	0.29	0.30	*p* = 0.34
** *S. strinatii* ** **Acarina**	**Parameter**	**Estimate**	**Std. Error**	***p*-value**
	Intercept	2.41	1.57	-
	*β*–Competitor’s presence (1)	0.29	2.71	*p* = 0.91
	*β*–SVL	−0.44	0.28	*p* = 0.11
	*β*–SVL * Competitor’s presence	−0.22	0.50	*p* = 0.66

The regression parameter estimate for the categorical variable (competitor’s presence) is measured as departures from the first level (competitor absent; 0). The size of the estimates, for each independent variable, gives the size of the effect that the variable has on resource use, and the sign of the estimate gives the direction of the effect. Significant effects, at alpha = 0.05, are marked with *.

## Data Availability

The data presented in this study are available from the corresponding author upon reasonable request.

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
