# Peer review of "Size-Mediated Trophic Interactions in Two Syntopic Forest Salamanders"

_animals, 2023, doi:10.3390/ani13081281_

Round 1
Reviewer 1 Report
Very nice research and great results! Compliments. Data are clearly presented. As table 3 is perhaps the most important table with figure 2 (in the headings of the figure it says 3 by the way) perhaps a bit more explaining in the results parts could be done. But it comes to life in the Discussion.
Author Response
We are grateful to the Reviewer for his enthusiastic review report. We made some change to the manuscript in the materials and methods and discussion sections to accommodate REV2 and REV3 requests.
Reviewer 2 Report
The idea of the manuscript is really interesting, and has a great potential in add information on the ecology of both species.
However, I don’t believe that the duration of the field work spent by the authors allow them to state broad findings. They sampled less than one month (how many days are not specified in the ms), which means they have only experienced a small variation in the environmental conditions, such as temperature, rains, the occurrence of other possible competitors and of prey occurrence. This prevents them to have any broader conclusion. This is clear an issue for these species since in the ms itself they state more than once that the studied species have different preferences on temperature when foraging.
So, I believe that to the manuscript to fully achieve their potential, the authors must sample in different months of the year, passing through different seasons, so they can decrease the chances of describing a brief pattern (maybe even a spurious one) as a general one.
Author Response
We are grateful with the Reviewer for his comments on our manuscript. We totally acknowledge his concern about the use of cross-sectional data on a single season and that this could be a limitation to the outcomes of the study. Therefore, following the suggestion of the Academic Editor, we made some change to the manuscript to better explain why we are confident on the results obtained. In particular, we changed the Materials and Methods section in order to stress out that the foraging strategy of the two study species, which has been extensively studied both in spring and autumn, is quite constant across seasons and only minimal changes occurs. Moreover, in the discussion section we acknowledge the issue raised by the Reviewer by adding a final paragraph, where we suggest that additional studies should be conducted using longitudinal data to better disclose the drivers of the interference observed. The paragraph is as follows:
“Finally, although the foraging strategy of the two study species is known to be consistent across seasons [34,43], our results were obtained from a cross-sectional study conducted only during spring. For this reason, we also suggest that future studies on this topic should consider to employ longitudinal diet data, instead of cross-sectional ones, in order to better disclose the magnitude of the interaction described, and how it could be modulated by seasonal effects, such as climatic conditions or resource availability.”
Reviewer 3 Report
Dear Authors,
I have reviewed your manuscript, and I am now submitting my revision. I really liked the manuscript, which I found interesting and answered the research question straight to the point. Indeed, I found some very minor issues needing to be addressed, which I list below:
L102 Maybe the reference [36] should be moved outside parentheses?
L158 Please add the R citation (R Core team)
L163 Please change to salamanders
L165 I understand that prey categories are prey taxa, but I think it would be better to specify it
Figure 3 Please add the meaning of colour bands in the figure caption. Also, a dashed line is reported in the plots, but a continuous one is then present in the legend.
L198 Please remove the doubled dot
L 334 and L 345 Italics missing to species’ names
My best regards,
Author Response
Authors: we are grateful to the Reviewer for his suggestions, which we followed modifying the manuscript accordingly:
L102 Maybe the reference [36] should be moved outside parentheses?
Authors:Changed as suggested
L158 Please add the R citation (R Core team)
Authors:Changed as suggested
L163 Please change to salamanders
Authors:Changed as suggested
L165 I understand that prey categories are prey taxa, but I think it would be better to specify it
Authors: we cknowledge reviewer’s comment, now it is specified parenthetically
Figure 3 Please add the meaning of colour bands in the figure caption. Also, a dashed line is reported in the plots, but a continuous one is then present in the legend.
Authors: we changed the figure legend as suggested and the figure caption states the meaning of colour bands as follows ”when the potential competitor is present (purple) or absent (green) on the forest floor”
L198 Please remove the doubled dot
Authors: removed as suggested
L 334 and L 345 Italics missing to species’ names
Authors: changed as suggested
Round 2
Reviewer 2 Report
I believe that the manuscript had an improvement regarding the clarification of important questions. After minor corrections, I believe that it will be ready to be published. Also, as far as I can judge, the English is good but can use some improvement.
